# A Pilot Randomized Controlled Trial of a Partial Meal Replacement Preconception Weight Loss Program for Women with Overweight and Obesity

**DOI:** 10.3390/nu13093200

**Published:** 2021-09-15

**Authors:** Roslyn Muirhead, Nathalie Kizirian, Ravin Lal, Kirsten Black, Ann Prys-Davies, Natasha Nassar, Louise Baur, Amanda Sainsbury, Arianne Sweeting, Tania Markovic, Michael Skilton, Jon Hyett, Bradley de Vries, William Tarnow-Mordi, Jennie Brand-Miller, Adrienne Gordon

**Affiliations:** 1Charles Perkins Centre, School of Life and Environmental Biosciences, University of Sydney, Sydney, NSW 2006, Australia; roslyn.muirhead@sydney.edu.au (R.M.); jennie.brandmiller@sydney.edu.au (J.B.-M.); 2Charles Perkins Centre, University of Sydney, Sydney, NSW 2006, Australia; nathalie.kizirian@sydney.edu.au (N.K.); ravin.lal@sydney.edu.au (R.L.); 3Speciality of Obstetrics, Gynaecology and Neonatology, Central Clinical School, Faculty of Medicine and Health, Charles Perkins Centre, University of Sydney, Sydney, NSW 2006, Australia; kirsten.black@sydney.edu.au; 4Department of Women and Babies, Royal Prince Alfred Hospital, Sydney Local Health District, Sydney, NSW 2050, Australia; apd@annprysdavies.com; 5Charles Perkins Centre, Childrens Hospital Westmead Clinical School, University of Sydney, Sydney, NSW 2006, Australia; natasha.nassar@sydney.edu.au (N.N.); louise.baur@sydney.edu.au (L.B.); 6School of Human Sciences, Faculty of Science, University of Western Australia, Crawley, WA 6009, Australia; amanda.salis@uwa.edu.au; 7Sydney Medical School, The University of Sydney, NSW 2006, Australia; Arianne.sweeting@sydney.edu.au (A.S.); tania.markovic@sydney.edu.au (T.M.); Michael.skilton@sydney.edu.au (M.S.); 8Charles Perkins Centre, Boden Initiative, The University of Sydney, NSW 2006, Australia; 9Department of Endocrinology, Royal Prince Alfred Hospital, NSW 2050, Australia; 10Speciality of Obstetrics, Gynaecology and Neonatology, Central Clinical School, Faculty of Medicine and Health, Royal Prince Alfred Hospital Newborn Care, Sydney Local Health District, Sydney, NSW 2050, Australia; jon.hyett@health.nsw.gov.au (J.H.); Bradley.devries@health.nsw.gov.au (B.d.V.); 11Clinical Trial Centre, Department of Neonatology, University of Sydney, Westmead Hospital, Sydney, NSW 2006, Australia; william.tarnow-mordi@sydney.edu.au; 12Sydney Medical School, Charles Perkins Centre, University of Sydney, Camperdown, NSW 2006, Australia; 13Royal Prince Alfred Hospital, Missenden Road, Camperdown, NSW 2050, Australia; 14Sydney Institute for Women, Children and their Families, Sydney Local Health District, Sydney, NSW 2000, Australia

**Keywords:** preconception, obesity, meal replacement, weight loss, clinical trial

## Abstract

About half of Australian women have a body mass index in the overweight or obese range at the start of pregnancy, with serious consequences including preterm birth, gestational hypertension and diabetes, caesarean section, stillbirth, and childhood obesity. Trials to limit weight gain during pregnancy have had limited success and reducing weight before pregnancy has greater potential to improve outcomes. The PreBabe Pilot study was a randomised controlled pilot trial to assess the feasibility, acceptability and potential weight loss achieved using a commercial online partial meal replacement program, (MR) vs. telephone-based conventional dietary advice, (DA) for pre-conception weight-loss over a 10-week period. Women 18–40 years of age with a BMI ≥ 25 kg/m^2^ planning pregnancy within the next 6 to 12 months were included in the study. All participants had three clinic visits with a dietitian and one obstetric consultation. In total, 50 women were enrolled in the study between June 2018 and October 2019–26 in MR and 24 in DA. Study retention at the end of 10 week intervention 81% in the MR arm and 75% in the DA arm. In the-intention-to-treat analysis, women using meal replacements lost on average 5.4 ± 3.1% body weight compared to 2.3 ± 4.2% for women receiving conventional advice (*p* = 0.029). Over 80% of women in the MR arm rated the support received as excellent, compared to 39% in the DA arm (*p* < 0.001). Women assigned to the MR intervention were more likely to achieve pregnancy within 12 months of the 10 week intervention (57% (12 of 21) women assigned to MR intervention vs. 22% (4 of 18) assigned to the DA group (*p* = 0.049) became pregnant). The findings suggest that a weight loss intervention using meal replacements in the preconception period was acceptable and may result in greater weight loss than conventional dietary advice alone.

## 1. Introduction

Australia is one of the most overweight developed nations, with two-thirds of adults and a quarter of children having overweight or obesity [1]. At a national level, women and men of reproductive age have the highest rate of weight gain [2]. As a result, approximately 50% of Australian women have overweight or obesity at the start of pregnancy with potentially serious consequences for both mother and baby [3]. Consequences include gestational diabetes (GDM) and maternal hypertension, preeclampsia, caesarean section and future cardiovascular disease [4]. Neonatal consequences include prematurity, stillbirth, congenital anomalies and higher birth weight (with possible birth injury). In addition to perinatal complications, in utero overnutrition is associated with adverse long-term consequences in the offspring, propagating an intergenerational cycle of obesity, diabetes and cardiovascular disease [5].

Despite the impact of obesity in pregnancy, systematic reviews of randomized controlled trials (RCTs) have concluded that interventions to limit or reduce weight gain in pregnancy do not succeed in substantially reducing risk of adverse pregnancy outcomes [6,7]. There is now increasing recognition that the period of pregnancy may be too late to address the risks of maternal obesity [8] and thus, preconception health might be key to improving perinatal health. The 2016 World Health Organization Commission report on Ending Childhood Obesity stressed the need for pre-conception interventions to improve the health of future generations, while others have emphasised early life as critical to intergenerational obesity, calling for promotion of interventions in preconception, inter-pregnancy and post-partum period to interrupt this cycle [9].

Previous studies in men and women with overweight and obesity have demonstrated that the amount of weight loss is greater with meal replacement products than with conventional dietary food restriction [6,7]. In a systematic review of RCTs and observational studies of women with infertility, 13 studies found greater weight loss with meal replacement regimes than with the other programs (9.4 ± 6.6 kg for a 12 week meal replacement program and 4.4 ± 5.8 kg for the largest diet/lifestyle trial) [10,11]. Meal replacement diets have also been shown to be more cost-effective in treating obesity and overweight in the short (6–12 months) and long-term (3–5 years) [12]. In women who have infertility as well as obesity, meal replacements substantially improved weight loss and pregnancy success [11,13]. However, the focus of these studies was conception and none reported longer term child outcomes beyond either pregnancy or a live birth.

Pre-conception weight loss is a key research priority in high income countries [5]. Although preconception weight management is recommended as a prevention strategy, there is a lack of randomised controlled trials to provide the necessary evidence to support recommendations [8]. 

The PreBabe Pilot study is a pilot RCT that aimed to assess the feasibility and acceptability of a partial meal replacement diet versus healthy diet advice for a duration of 10 weeks in women with a BMI ≥ 25 kg/m^2^ who were planning a pregnancy within the next 6 to 12 months. We hypothesised that women would achieve greater weight loss on the partial meal replacement diet compared with standardised healthy dietary advice, and that the program would be acceptable to the women and feasible to scale-up for a larger RCT.

## 2. Materials and Methods

The trial was registered with the Australian New Zealand Clinical Trials Registry (http://www.anzctr.org.au ACTRN12620000597998; Date Registered: 22 May 2020) and approved on 15 February 2018 by the Sydney Local Health District RPA Zone Human Research Ethics Committee (Protocol X17-0382 & HREC/17/RPAH/579).

### 2.1. Participants 

Women with a BMI ≥ 25 kg/m^2^ living in the Sydney Local Health District (SLHD), New South Wales, Australia, who were planning a pregnancy within 6 to 12 months were eligible to participate. Inclusion criteria were: aged between 18 and 40 years inclusive; Body Mass Index (BMI) ≥ 25 kg/m^2^ from measured height and weight; intending a pregnancy within the next 6 to 12 months; being weight stable (i.e., <3 kg weight loss/gain) in the past 2 months and willing and able to attend the Charles Perkins Centre- Royal Prince Alfred Hospital Clinic (RPAH), University of Sydney on three occasions. Exclusion criteria were BMI < 25 kg/m^2^; currently pregnant, breastfeeding or <6 months postpartum; currently taking weight loss medication; diagnosed pre-existing medical condition that is contra-indicated for a weight loss study, e.g., Type 1 diabetes, severe depression, malignancy, previous weight loss surgery. Recruitment strategies included: flyer distribution within the Sydney Local Health District including the pregnancy planning clinic, antenatal, gestational diabetes post-partum and fertility clinics; local medical centres and pharmacies; and local childcare/playgroup facilities. Both paid and free study advertisements were distributed via pregnancy social media and targeted websites, as well asa webpage within the University of Sydney with optimised Google Search and newspaper articles. The study team also gave presentations to primary health care networks aligned with the Sydney Local Health District. Recruitment material directed women to the study team via a research telephone or email account. Interested women were sent the participant information sheet and study flyer and completed an online questionnaire assessing study eligibility. Women meeting eligibility criteria were then contacted by the study team to arrange consent, randomisation and baseline assessment. Women provided signed informed consent prior to data collection and were able to withdraw from the trial at any time. Recruitment commenced in June 2018 and finished in December 2019. Follow up of conception and pregnancy continued until December 2020. 

### 2.2. Study Design

This was a single site randomised controlled trial. Eligible women were randomised to a 10-week protocol of either a partial meal replacement diet (Flexi by Impromy™, Blackmores, Warriewood, NSW Australia) [14] or to the Get Healthy telephone coaching service (New South Wales Department of Health) for recommended healthy dietary advice [15]. Both arms received three face-to-face visits with a trained research dietitianand a consultation with an obstetrician at their first study visit. Women were randomised in a 1:1 ratio using a variable block randomisation sequence generated by computer software (sealedenvelope™ Sealed Envelope Ltd, Clerkenwell Workshops, London, UK). Randomisation was stratified for BMI 25–29.99 kg/m^2^ and BMI ≥ 30 kg/m^2^, and undertaken by staff not involved in the intervention. Staff responsible for data analyses were also blinded, but the nature of the intervention meant the participants and the research dietitian could not be blinded. 

### 2.3. Study Visits

Protocol timeline and data collection are shown in Table 1. At the baseline visit 1 (week 1), the research team confirmed eligibility and participants gave written informed consent. Women had their height, weight and waist measured, BMI calculated and a venous blood sample drawn for baseline biochemistry. In addition, women completed a study entry questionnaire, including the Depression, Anxiety and Stress Scale-21 (DASS-21) [16] and the Australian Eating Survey, an online food frequency questionnaire [17].

At visit 2 (week 5), usual dietary intake and physical activity were reviewed by the dietitian. Participants were weighed and received personalised dietary recommendations from the dietitian based on the assigned intervention and the results generated by the Australian Eating Survey. At visit 3 (week 10), women completed anthropometric assessments and a repeat blood sample was drawn. Four to 6 weeks after visit 3, participants were asked to complete the online Australian Eating Survey for a second time to assess post-intervention diet. Thereafter, participants were contacted at monthly intervals for 12 months to collect data on conception and pregnancy. 

### 2.4. Partial Meal Replacement Protocol

Participants randomised to the meal replacement (MR) intervention group followed a partial meal replacement diet for 10 weeks, using shakes (liquid beverages) provided by Blackmores Australia. The program (Figure 1) was developed in collaboration with CSIRO (the Commonwealth Scientific and Industrial Research Organisation) and previously shown to be effective in adults with overweight or obesity [18]. For 6 days each week, energy intake was restricted with alternating Classic Days and Control Days. On Classic Days, participants consumed a specific number of meal replacement shakes + low energy snacks and a higher protein meal. The prescription for energy intake was 30% less than estimated energy requirements. On Control Days, they consumed only meal replacement shakes + a low-energy vegetable meal (around 45–50% of estimated energy requirements). Guidelines and recipes for preparing meals and permitted snacks were provided. Permitted snacks consisted of 500 kJ portions of fruit, low-fat dairy, wholegrains, and nut/seed/legume options. On one day each week, participants were permitted *ad libitum* eating. Individual estimated energy requirements were calculated at study commencement based on body weight and then reduced by 30% to determine the exact number of permitted sachets per week. Goal weight loss was set at 5% of starting body weight. 

Meal replacements were reconstituted with 250 mL of either skim milk or dairy-free alternative (unsweetened, calcium-enriched). The nutrient composition as consumed was therefore ~1000 kJ, 25 g protein, 4 g fat, 27 g carbohydrate, and 6 g fiber, with each containing 25% recommended daily intake for vitamin A, thiamin, riboflavin, niacin, folate, vitamin B6, vitamin B12, vitamin C, vitamin D, vitamin E, calcium, iodine, iron, magnesium, phosphorus, and zinc [14]. Participants were encouraged to drink fluids (25–35 mL/kg^−1^ body weight/day^−1^) and advised of optional low energy beverages, vegetables and condiments with minimal kilojoule content that could be consumed as needed to manage hunger. On the *ad libitum* day (one day a week), women were advised that they could eat to appetite, choosing the type and quantity of food and beverages they desired (intake was not recorded). Exercise advice was based on Australian and World Health Organisation guidelines (accumulate 150–300 mins of moderate intensity physical activity each week). As this was not a total meal replacement diet, a refeeding phase was not considered necessary.

### 2.5. Conventional Dietary Advice Protocol

Participants in the conventional dietary advice (DA) arm received healthy diet advice via the Get Healthy^®^ Information & Coaching Service provided by the New South Wales Department of Health, Australia [15]. This government-sponsored service is a freely available one-on-one weight loss program with telephone support delivered by a health coach. The program is delivered in 6 stages, with goal-setting, food intake review, strategic advice (e.g., overcoming roadblocks) and weight checks. 

### 2.6. Data Collection and Management

Demographic characteristics were collected via questionnaire at baseline, including age, education, occupation (day shift/night shift), marital status, ethnicity, medical history, previous pregnancies. Weight was measured in light clothing and without shoes on a digital scale (Wedderburn BC-418). Height was measured in metres to the nearest millimeter using a Holtain Harpenden stadiometer (Holtain Ltd., Crymych, UK). Waist circumference (cm) was measured midway between the bottom of the ribcage and the top of the iliac crest using a metal tape measure. 

Dietary intake was assessed using the Australian Eating Survey^®^ (Version 10), a validated Food Frequency Questionnaire [17]. The intervention was assessed by the participants 4 weeks after the final visit via an online questionnaire. Pregnancies were confirmed at 3 monthly intervals up to 12 months post-intervention. Acceptability was assessed by the evaluation questionnaire. Feasibility was judged by recruitment and retention in the pilot trial. Data were entered into the REDCap^®^ research management system (REDCap 8.2.2 ^©^ Vanderbilt University, 2201 West End Ave, Nashville, TN 37235, USA), a secure web application supported by the University of Sydney and Sydney Local Health District.

### 2.7. Sample Size

The purpose of the pilot study was to assess feasibility and acceptability of the partial meal replacement diet intervention, and therefore not powered on clinical outcomes. A total of 50 women was considered adequate to provide realistic feedback, recruitment, compliance and retention as well as average weight loss per arm. The findings will inform a larger pragmatic trial designed to assess the effect of weight loss preconception on clinical, maternal and newborn outcomes.

### 2.8. Statistical Treatment

Data are presented as mean ± SD, unless otherwise indicated. Independent samples t-test was used to determine equality of means A repeated measures ANOVA with a Greenhouse-Geisser correction was also used to determine within-group differences in mean body weight and waist circumference from baseline and week 10 (MR vs. DA). Pearson chi-square test of independence was used to determine differences in categorical outcomes, including completion rate. Fishers exact test was used where cell sizes were less than 5. An intention-to-treat analysis was used with missing values considered missing at random. 

## 3. Results

Flow of participants through the study is shown in Figure 2. In total 132 women were assessed for eligibility. Of these, 40 did not meet the inclusion criteria, 39 declined the invitation to participate and 3 were excluded for other reasons. Fifty women were randomized, 26 to MR and 24 to the DA group. Five women in the MR and 6 in the DA arm withdrew early (after visit 1 and prior to visit 2). Baseline characteristics were similar in MR vs. DA, including age (33.7 (±3.8) and 31.5 (±6.9) years), BMI (34.7(±5.9) vs. 32.9 (±5.9) kg/m^2^), education, marital status, occupation, medical history, parity and any previous pregnancy loss (miscarriage or stillbirth) (Table 2). On a scale from 0 to 21, DASS-21 scores were also comparable, averaging 6–7, 5–6, 10–11 on depression, anxiety and stress scales respectively (Table 2). 

Study retention was not statistically different between groups: 81% (21 of 26) in the MR arm vs. 75% (18 of 24) in the DA group (*p* = 0.623). In the intention-to-treat analysis, women in MR lost (mean ± SD) 5.3 ± 3.9% of body weight compared to 2.8 ± 3.9% in the DA group (*p* = 0.029, Table 3). There was no difference in waist circumference: 5.3 ± 4.4 cm vs. 5.4 ± 4.8 cm in MR vs. DA group respectively, *p* = 0.941.

Overall, there were no differences between the groups in markers of glycemia (HbA_1c_) or lipidemia (total cholesterol, HDL, LDL), either at baseline or at the end of the intervention (week 10) (Table 4). Triglycerides were lower in the MR vs. DA arm at Week 10 however there was no significant difference in the change over time between the groups. Nutritional status (iron status, serum folate or vitamin D improved on both programs with no differences between the groups (Table 4). 

There were differences in the women’s ratings of the quality of support (Appendix A). Over 80% if those in the MR arm rated the support they received as excellent, compared to only 39% of those in the DA arm (*p* < 0.001). In the evaluation survey 95% of the women assigned MR would recommend the program to a friend compared with 58% in the DA arm (*p* < 0.001). There were no significant differences in dietary intake over time for either group other than percentage of energy eaten as protein (Appendix A). 57% (12 of 21) women on meal replacements became pregnant within 12 months of the 10 week timepoint vs. 22% (4 of 18) in the DA group (*p* = 0.049).

## 4. Discussion

The aim of this pilot randomized controlled trial was to explore the feasibility and acceptability of a partial meal replacement program in the preconception period for women with overweight or obesity and the potential weight loss acheived. Our findings indicate that a commercially available online partial meal replacement program was feasible in that it was both achievable and highly acceptable to women with overweight or obesity who were planning a pregnancy. Average weight loss was 1.9 times higher in the meal replacement arm (5.3%) than conventional advice group (2.8%, *p* = 0.029). Retention was similar in both groups (81% and 75% respectively) and within normal bounds for nutrition intervention studies in this population [18]. Women in the MR program also rated the online support structure more highly than the women receiving the telephone advice offered to the DA group. The findings in this pilot are intended to inform the planning of a larger, appropriately powered trial where a composite measure of maternal and infant outcomes will be the primary outcome.

The finding that a weight loss program is both feasible and acceptable to women with overweight and obesity planning pregnancy and that greater weight loss can be achieved through the use of meal replacements is consistent with prior data in adult populations [6,7] and an important addition to the literature for women planning pregnancy. There is increasing recognition that pregnancy is not the best time to address maternal obesity [19] with guidelines proposing pre-conception management of obesity as a strategy to prevent adverse perinatal consequences, yet no evidence from randomised trials to support this recommendation [13]. Prior weight loss trials in the pre-pregnancy period have mainly focused on women with known sub-fertility [9,20,21,22] whereas interventions in pregnancy focus on limiting weight gain rather than achieving weight loss. Strategies only focusing on limiting weight gain during pregnancy do not appear to result in the anticipated benefit in clinical outcomes for mother or child [23,24,25]. The largest RCT to test limiting gestational weight gain to date is the LIMIT trial, performed in Australia which assessed a lifestyle intervention (diet and physical activity) in pregnant women with overweight or obesity (*n* = 2212) [23]. This well designed and adequately powered trial did not demonstrate any significant risk reduction in the incidence of the primary outcome (i.e., risk of large-for-gestational age infant (LGA), from 14.4% to 10.1%) and there was no difference in gestational weight gain for the women. However, LIMIT did show an 18% relative risk reduction (from 19% to 14%) in the secondary outcome of macrosomia (birth weight >4 kg) in the intervention arm compared to control. Similar findings were seen in the UPBEAT Trial performed in the UK (*n* = 1556). Despite improvements in diet, there was minimal change in gestational weight gain (mean difference 0.55 kg) and no demonstrable impact on clinical outcomes for the mother (including gestational diabetes and pre-eclampsia) or the baby (including LGA and hospital admission) [26]. These findings may relate to the fact that, inevitably, any dietary intervention in pregnancy cannot be as intensive as outside of pregnancy, such as in the pre-conception period. As weight loss in pregnancy is not considered safe [27], RCTs before conception are critically important.

The strengths of this pilot study include its randomised, controlled trial design and the inclusion of women with overweight or obesity in the general population rather than those already seeking treatment through fertility services. We ran the trial through existing clinical services to ensure it was translatable to clinical settings. The trial included face-to-face collection of critical data by trained individuals, informed all participants about preconception health and had the same number of visits with a professional in both groups [28]. We successfully targeted women who may not have otherwise sought pre-conception weight loss, i.e., a group that is not usually part of the health system to provide a service that would be cost-effective and accessible in the long term. Limitations to this pilot trial include the fact that those who participated were a highly educated cohort, most having a tertiary qualification and that recruitment took longer than anticipated. This was primarily due to limited funding for advertising and the access to the clinic only one morning per week, underlining the challenges of undertaking a large-scale intensive dietary intervention in this group and the importance of allocating funding specifically towards recruitment strategies.

## 5. Conclusions

This pilot study in 50 women provides the evidence of feasibility and acceptability of a commercially available meal replacement weight loss program for women with overweight and obesity planning a pregnancy. Further, the program was more acceptable to women compared to currently available phone counselling options. The findings are relevant to the planning of larger trials of safety and efficacy. 

## Figures and Tables

**Figure 1 nutrients-13-03200-f001:**
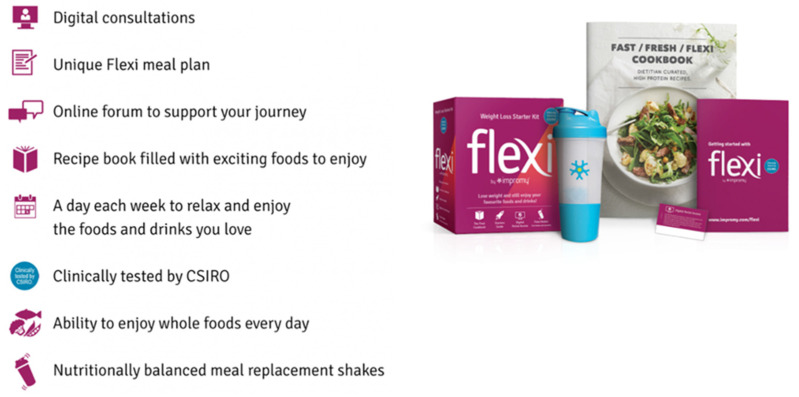
Partial Meal Replacement Program.

**Figure 2 nutrients-13-03200-f002:**
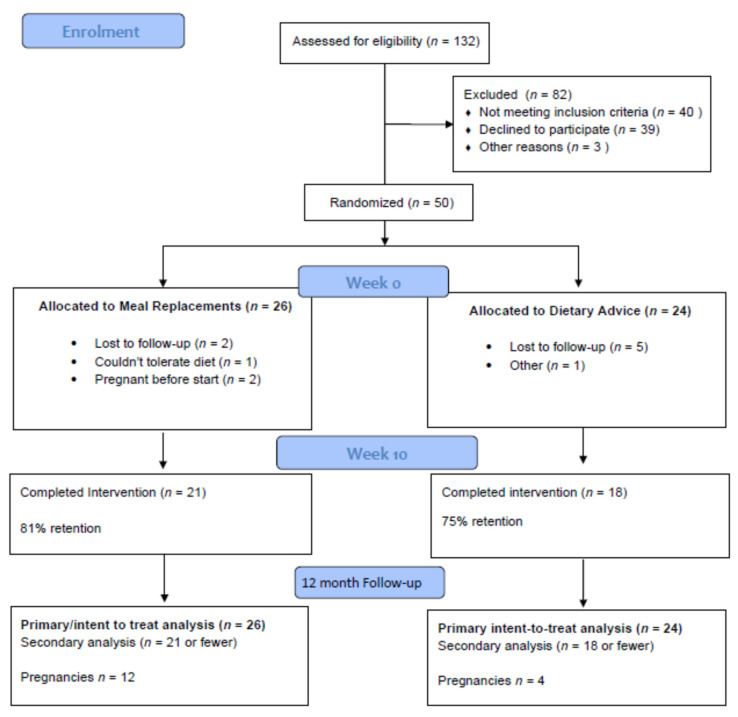
Flow of participants through the study.

**Table 1 nutrients-13-03200-t001:** Protocol and Data Collection Timeline.

	Screening	Week 1 Visit 1	Week 5 Visit 2	Week 10 Visit 3	Week 10 +1 m	Week 10 to 12 Months
Screen	X					
Consent		X				
Allocation		X				
Clinic visit		X	X	X		
Anthropometry		X		X		
Venous blood		X		X		
Questionnaire		X				
Depression, Anxiety and Stress Scale 21DASS-21		X				
Dietary survey		X			X	
Dietary advice		X				
Diet review			X			
Evaluation					X	
Conception						Quarterly text/email

**Table 2 nutrients-13-03200-t002:** Baseline participant characteristics in the two intervention groups.

	Dietary Advice (*n* = 24)	Meal Replacements (*n* = 26)
Age (years)	33.7 ± 3.8	31.5 ± 6.9
Weight (kg)	92.0 ± 16.2	89.5 ± 17.1
Height (cm)	162.8 ± 5.9	164.9 ± 6.1
Body Mass Index (kg/m^2^)	34.7 ± 5.9	32.9 ± 5.9
Waist circumference (cm)	101.7 ± 11.7	97.9 ± 11.1
Education: Secondary *n* (%)	3 (13%)	1 (4%)
Tertiary	11 (46%)	12 (46%)
Post graduate degree	10 (42%)	13 (50%)
Occupation: Do not have a job (*n*)	0	2
Part-time	9	8
Full time	14	15
Student or other	1	1
Medical history: Hypertension (*n*)	2	0
High cholesterol (*n*)	2	1
Pre-diabetes or gestational diabetes (*n*)	3	2
Obesity (*n*)	7	4
Polycystic ovary syndrome (*n*)	1	2
Other conditions *	11	7
Number of prior pregnancies: 0	12	10
1	7	13
2 or more	5	3
Number of children: 0	16	15
1	7	11
2 or more	2	0
Previous pregnancy loss:Yes (*n*)	6	5
No (*n*)	18	21
Depression (DASS-21)	6.1 ± 6.4	6.8 ± 5.7
Anxiety (DASS-21)	6.8 ± 6.4	5.4 ± 5.6
Stress (DASS-21)	10.9 ± 8.3	10.5 ± 7.0

Values are Mean ± SD, and number of participants (*n*). No statistically significant differences between groups at baseline. * Includes thyroid disease, congential heart disorder, coronary bypass, genetic, Crohn’s disease, asthma and anxiety/depression. Note medical conditions are not mutually exclusive—a participant could have more than one condition.

**Table 3 nutrients-13-03200-t003:** Changes in weight and waist circumference.

*n*	Dietary Advice		Meal Replacements		*p*-Value ^1^
	Baseline	Week 10	Baseline	Week 10	
	24	18	26	21	
Weight (kg)	92.0 ± 16.2	86.3 ± 13.6	89.5 ± 17.1	84.9 ± 16.8	0.949
Weight loss (kg)		2.7 ± 3.8		4.7 ± 2.8	0.061
Weight loss (%)		2.8 ± 3.9		5.3 ± 3.0	0.029
Waist (cm)	101.7 ± 11.7	94.3 ± 9.0	97.9 ± 11.1	93.9 ± 11.7	0.900
Waist loss (cm)		5.4 ± 4.8		5.3 ± 4.4	0.941
Waist loss (%)		5.7 ± 5.2		5.6 ± 4.9	0.912

Values are mean ± SD. ^1^ *p*-values are reported for between-group differences at 10 weeks.

**Table 4 nutrients-13-03200-t004:** Changes in blood chemistry status in the Dietary Advice and Meal Replacement groups before and after the 10-week intervention.

	Dietary Advice		Mean Difference	Meal Replacements		Mean Difference
	Baseline	Week 10		Baseline	Week 10	
HbA1c (mmol/mol)	32.55 ± 4.17 (*n* = 20)	33.07 ± 4.60 (*n* = 14)	0 ± 1.8	30.89 ± 6.13 (*n* = 26)	31.16 ± 2.39 (*n* = 19)	0.15± 6.6
Total cholesterol (mmol/L)	4.84 ± 1.07 (*n* = 20)	5.00 ± 0.99 (*n* = 15)	0.04 ± 0.9	4.88 ± 0.89 (*n* = 26)	4.51 ± 0.97 (*n* = 18)	0.33 ± 0.6
Triglycerides (mmol/L)	1.84 ± 1.13 (*n* = 21)	1.96 ± 1.58 (*n* = 15)	0.14 ± 1.5	1.31 ± 0.63 (*n* = 25)	1.17 ± 0.49 (*n* = 19)	0.13 ± 0.5
Iron (umol/L)	14.70 ± 4.61 (*n* = 20)	16.73 ± 5.74 (*n* = 15)	0.92 ± 5.7	14.50 ± 4.53 (*n* = 26)	14.95 ± 4.90 (*n* = 19)	0.53 ± 4.1
Ferritin (ug/L)	77.40 ± 52.56 (*n* = 20)	84.27 ± 54.58 (*n* = 15)	5.85 ± 39.5	81.38 ± 59.97 (*n* = 26)	75.84 ± 49.93 (*n* = 19)	6.68 ± 26.1
Transferrin (g/L)	2.83 ± 0.36 (*n* = 20)	2.80 ± 0.40 (*n* = 15)	0.12 ± 0.31	2.77 ± 0.42 (*n* = 26)	2.80 ± 0.34 (*n* = 19)	0.053 ± 0.18
Transferrin saturation (%)	20.60 ± 5.77 (*n* = 20)	24.53 ± 10.50 (*n* = 15)	3.00 ± 10.2	21.42 ± 7.57 (*n* = 26)	21.32 ± 7.16 (*n* = 19)	0.74 ± 5.9
Serum folate (nmol/L)	36.12 ± 9.53 (*n* = 20)	40.30 ± 7.62 (*n* = 15)	3.33 ± 6.0	35.03 ± 9.11 (*n* = 26)	35.93 ± 8.55 (*n* = 19)	0.98 ± 9.7
Vitamin D (nmol/L)	61.76 ± 24.86 (*n* = 21)	66.53 ± 24.91 (*n* = 15)	9.0 ± 27.5	65.04 ± 25.24 (*n* = 24)	70.00 ± 26.11 (*n* = 19)	3.53 ± 19.1

Values are mean ± SD. There were no significant differences in the changes between baseline and Week 10 for any of the blood tests in either randomised group or between groups.

## Data Availability

The data presented in this study are available on request from the corresponding author.

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
