# Peer review of "A Pilot Randomized Controlled Trial of a Partial Meal Replacement Preconception Weight Loss Program for Women with Overweight and Obesity"

_nutrients, 2021, doi:10.3390/nu13093200_

Round 1

Reviewer 1 Report

Thank you very much for the presentation of this trial. Please find enclosed my comments:

page 1 line 51 Please check the brackets.

Page 2  The authors stated that the follow up  ended in December 2020. Have the authors planned to receive data about the ongoining of the pregnancy and children after birth?

Page 4 A figure of the MR would be helpful.

Page 4 How was the energy requirement calculated? Based on the energy requirement for the BMI of <25 kg/m²?

Author Response

1) page 1 line 51 Please check the brackets.

Corrected

2) Page 2  The authors stated that the follow up  ended in December 2020. Have the authors planned to receive data about the ongoining of the pregnancy and children after birth?

We have not continued follow up beyond 12 months post intervention for this pilot study which was assessing feasibility however would aim to do this in any larger trials powered to assess clinical outcomes

3)Page 4 A figure of the MR would be helpful.

Figure 2 added

4)Page 4 How was the energy requirement calculated? Based on the energy requirement for the BMI of <25 kg/m²?

Already stated on page 4: Individual estimated energy requirements were calculated at study commencement based on body weight and then reduced by 30% to determine the exact number of permitted sachets per week

Reviewer 2 Report

Comments and Suggestions for Authors

The study by Muirhead et al. aimed to assess the feasibility, acceptability, and potential weight loss achieved using a meal replacement program among non-pregnant women with excessive weight.

This pilot intervention study presents a very important area in clinical practice.  Because there are more and more overweight or obese women who want to be pregnant and the recommendations of the most effective weight loss in this group of patients are still not known, as well as the feasibility and acceptability of a partial meal replacement diet.

My minor comments related to the study are as below:

Title:

The title is confusing and not fully related to the aim of the study.

The authors would like to underline only "the feasibility" of the weight loss program?

Abstract:

Page 1, line 51: please complete the missing brackets

1.Introduction:

The introduction is well designed.

2.Materials and Methods

Page 3, line 115: remove one of the double used words “was”

Page 3, lines 148-149: remove one of two dots.

Page 5, lines 196-197: add forward slash in the bracket

Page 5, line 216: change a capital letter in the word “To” into lower case  

3.Results

Page 6, table 2: add p-value to the participant's characteristics presented in Table 2

Page 8, line 273: remove one of two dots

4.Discussion

Page 9, line 311: Lower case in the word “despite” change into a capital letter.

The discussion section and conclusions must be improved by structuring the discussion of the findings more clearly.

The authors have not included a thorough discussion of previous studies in the area. It would help the reader’s understanding of where the results of this study sit in this literature by more clearly stating the results that were found in this study at the beginning of each paragraph in the discussion section and then discussing how this finding fits with the evidence from previous studies.

Author Response

Title:

The title is confusing and not fully related to the aim of the study.

Thank you – title reworded

Abstract:

Page 1, line 51: please complete the missing brackets

Done

1.Introduction:

The introduction is well designed.

2.Materials and Methods

Page 3, line 115: remove one of the double used words “was” - completed

Page 3, lines 148-149: remove one of two dots - completed

Page 5, lines 196-197: add forward slash in the bracket - completed

Page 5, line 216: change a capital letter in the word “To” into lower case - completed

3.Results

Page 6, table 2: add p-value to the participant's characteristics presented in Table 2

The differences are non significant - noted already in the existing foot note for Table 2 , hence why the p values are not presented

Page 8, line 273: remove one of two dots - completed

4.Discussion

Page 9, line 311: Lower case in the word “despite” change into a capital letter - completed

The discussion section and conclusions must be improved by structuring the discussion of the findings more clearly.

We have expanded on the discussion and hope this is now clearer

Reviewer 3 Report

Review of nutrients-1350364

General comments: This is a paper assessing the feasibility of two different weight loss programs; one, a telephone-based conventional dietary program, and the other, a commercial online partial meal replacement program. The two different programs were carried out with three dietetic sessions and an assessment of anthropometry at the start, mid, and end of the study. The meal replacement arm was superior in relative weight loss (%) but not in absolute weight loss or waist circumference. At the following evaluation, the meal replacement arm had higher satisfaction of support.

Generally, the manuscript is well written and easily understandable, but there are some issues that need improvement.

The authors used intention to treat analyses as a single comparison between groups. Available case or even better completers analyses would be more appropriate to assess the effectiveness of the used method. Even though being a feasibility study, the ability to create a weight loss cannot be neglected.

Also, include a discussion of the very long time of recruitment and what impact this may have, i.e. was it influenced by the type of intervention?

Specific comments:

Line 52: include an ‘a’: “…weight loss over a 10-week period”

Line 52: include ‘between’: “Women between 18-40 years”

Line 140-141: Did the women not receive oral information?

Line 164-166: Were all anthropometric measurements performed at vs 2; only weight is stated in the text. Please align text and table.

Line 184: What was the energy intake/restriction during control days?

Line 216: Include manufacturer name and country.

Line 216: Replace capital T with small ‘t’

Line 242: Perform data analyses using available case and complete case analyses.

Line 249: If women withdrew before starting it is not possible to conclude upon acceptability based on retention rate as otherwise stated in line 224.

Line 250: Include SD in both BMI and age

Figure 1: I the box “Allocated to meal replacement” there seems to be an extra ‘lost to follow-up’; ensure this is correct. Moreover, please optimize exchanging the two arms, so it is similar to presented in the tables

Table 2: delete “Mean” in the variables; it is described in the table text and therefore does not have to be repeated. In “number of pregnancies” combine 2, 3, and 4 or more as there are so few. Similar in “number of children” combine 2 and 3.

Line 262: Show data from the complete and available case analyses.

Line 263-264: Include differences between groups instead of repeating data from the table in the text.

Table 3: Include results in waist percentage

Table 4: Different group names are used in the title; please align. Include differences between groups. The table is difficult to read with the current setup, please improve. Present 95%CI for differences.

Line 311: Capitalize “Despite”

Line 321: Please explain the focus on clinical translation

Line 325-326: How do you know you have recruited women who not otherwise would have sought weight loss?

Line 335: you write “may show” in the conclusion please describe what the evidence from the study results show and not what it might.

Line 336: Are the age and socioeconomic classes different and how do you know this? You highlight the women being in the general population in line 319.

Author Response

The authors used intention to treat analyses as a single comparison between groups. Available case or even better completers analyses would be more appropriate to assess the effectiveness of the used method. Even though being a feasibility study, the ability to create a weight loss cannot be neglected. – We understand these issues but are not convinced that this additional analysis will value add to what is essentially a pilot feasibility study

Also, include a discussion of the very long time of recruitment and what impact this may have, i.e. was it influenced by the type of intervention? – added to discussion

Specific comments:

Line 52: include an ‘a’: “…weight loss over a 10-week period” - completed

Line 52: include ‘between’: “Women between 18-40 years” – we have not changed this as believe it then can read like 18  and 40 are not included so have left this as submitted

Line 140-141: Did the women not receive oral information? – Yes they did at the documented in person visits

Line 164-166: Were all anthropometric measurements performed at vs 2; only weight is stated in the text. Please align text and table. – Only weight at visit 2 and full anthropomentry at baseline and 10 weeks – table altered

Line 184: What was the energy intake/restriction during control days? Added to the text for control days

Line 216: Include manufacturer name and country. - completed

Line 216: Replace capital T with small ‘t’ - completed

Line 242: Perform data analyses using available case and complete case analyses.

Thank you for this suggestion. Following discussion our team don’t feel this additional analysis will significantly change the results or add to the paper. The study is a small pilot study performed mainly to assess feasibility and acceptability in a target population to inform larger trials. If a participant did not complete we do not have their week 10 weight available.

Line 249: If women withdrew before starting it is not possible to conclude upon acceptability based on retention rate as otherwise stated in line 224. –

Yes agree this is unclear. These withdrawal occurred early (after visit 1 and before visit 5) with some having commenced their allocated intervention and some not -  We have reworded accordingly.

Line 250: Include SD in both BMI and age - completed

Figure 1: I the box “Allocated to meal replacement” there seems to be an extra ‘lost to follow-up’; ensure this is correct. Moreover, please optimize exchanging the two arms, so it is similar to presented in the tables – This error has been corrected

Table 2: delete “Mean” in the variables; it is described in the table text and therefore does not have to be repeated. In “number of pregnancies” combine 2, 3, and 4 or more as there are so few. Similar in “number of children” combine 2 and 3 - completed.

Line 262: Show data from the complete and available case analyses - Refer to earlier comment

Line 263-264: Include differences between groups instead of repeating data from the table in the text. – have reworded

Table 3: Include results in waist percentage - completed

Table 4: Different group names are used in the title; please align. Include differences between groups. The table is difficult to read with the current setup, please improve. Present 95%CI for differences. – Table 4 amended

Line 311: Capitalize “Despite” - complete

Line 321: Please explain the focus on clinical translation - completed

Line 325-326: How do you know you have recruited women who not otherwise would have sought weight loss?  

This sentence is to explain that such women are often otherwise healthy and not engaged in a health system. We have expanded to make this clearer as below:

“We successfully targeted women who may not have otherwise sought pre-conception weight loss, ie, a group that is generally healthy and therefore not usually part of the health system”

Line 335: you write “may show” in the conclusion please describe what the evidence from the study results show and not what it might. – altered accordingly

Line 336: Are the age and socioeconomic classes different and how do you know this? You highlight the women being in the general population in line 319. – they are not different, we were just stating that for the women we assessed in this trial the program was acceptable – we have altered accordingly

Round 2

Reviewer 3 Report

Thank you for the performed amendments.

In the text and tables, standard deviations are presented both as (sd xx) and ± xx; please unify using either one or the other and if you use the parentheses it's capital SD. 

Author Response

in the text and tables, standard deviations are presented both as (sd xx) and ± xx; please unify using either one or the other and if you use the parentheses it's capital SD.

Thank you - have unified throughout text and tables to ± xx